# Altertoxins with Quorum Sensing Inhibitory Activities from The Marine-Derived Fungus *Cladosporium* sp. KFD33

**DOI:** 10.3390/md18010067

**Published:** 2020-01-19

**Authors:** Fei Zhang, Liman Zhou, Fandong Kong, Qingyun Ma, Qingyi Xie, Jiuhui Li, Haofu Dai, Lei Guo, Youxing Zhao

**Affiliations:** 1Hainan Key Laboratory of Research and Development of Natural Product from Li Folk Medicine, Institute of Tropical Bioscience and Biotechnology, Chinese Academy of Tropical Agricultural Sciences, Haikou 571101, China; MarchFay@163.com (F.Z.); zhouliman88@126.com (L.Z.); kongfandong@itbb.org.cn (F.K.); maqingyun@itbb.org.cn (Q.M.); xieqingyi@itbb.org.cn (Q.X.); lijiuhui@itbb.org.cn (J.L.); daihaofu@itbb.org.cn (H.D.); 2Jiangsu Key Laboratory of Marine Bioresources and Environment, Co-Innovation Center of Jiangsu Marine Bio-industry Technology, Jiangsu Ocean University, Lianyungang 222005, China

**Keywords:** *Cladosporium* sp., altertoxins, quorum sensing inhibitory activity

## Abstract

Five new perylenequinone derivatives, altertoxins VIII–XII (**1**–**5**), as well as one known compound cladosporol I (**6**), were isolated from the fermentation broth of the marine-derived fungus *Cladosporium* sp. KFD33 from a blood cockle from Haikou Bay, China. Their structures were determined based on spectroscopic methods and ECD spectra analysis along with quantum ECD calculations. Compounds **1**–**6** exhibited quorum sensing inhibitory activities against *Chromobacterium violaceum* CV026 with MIC values of 30, 30, 20, 30, 20 and 30 μg/well, respectively.

## 1. Introduction

The unremitting discovery of antibiotics showed a promising prospect for the bacterial infections, but also resulted in the serious resistance of antibiotics. Bacteria have evolved a variety of resistance mechanisms [1,2]. Quorum sensing (QS) is the regulation of gene expression in response to fluctuations in cell-population density among diverse bacterial species [3]. QS controls the production of virulence factors in bacteria in a population density dependent manner through intercellular communication mechanism [3]. QS inhibitors (QSIs) can inhibit the QS mechanism and attenuate virulence without influencing bacterial growth. Thus, QSIs can be used to disarm pathogens in the host and are not easy to cause bacterial resistance compared to conventional antibiotics [4,5]. The search for efficient QSIs is supposed to be an effective method to solve problems of bacterial infection and antibiotic resistance. Based on this, a screening system has been established for searching of QSIs [6]. Due to the special environmental conditions, marine-derived fungi, as a rich source of various compounds with complex structures and excellent activities, have attracted more and more attentions [7]. Our previous research on new bioactive metabolites from the marine-derived fungi has led to the isolation and identification of many new QSIs, such as aculene E and penicitor B, aculene C, aculene D, aspergillumarins A–B [8], asperochrin D, asperochrin F, (3*R*,4*R*)-4,7-dihydroxymellein, asperochrin A, and asteltoxin [9]. As our ongoing research, the fungus *Cladosporium* sp. KFD33 was isolated from a blood cockle from Haikou Bay, China. The EtOAc extract of the fermentation broth of this fungus showed obvious QS inhibitory activity against *Chromobacterium violaceum* CV026. Subsequent chemical investigation on the EtOAc extract of the fermentation broth had led to the isolation of five new compounds, named altertoxins VIII–XII (**1**–**5**), as well as a known one, cladosporol I (**6**) [10] (Figure 1). All of the new compounds showed obvious QS inhibitory activities. Herein, the isolation, structure elucidation, and QS inhibitory activity of compounds **1**–**6** are described. 

## 2. Results and Discussions

Compound **1** was obtained as a dark yellow powder, and its molecular formula was determined as C_20_H_16_O_3_ on the basis of HRESIMS data, indicating 13 degrees of unsaturation. The ^13^C NMR and HSQC spectra showed 20 carbon signals assigned to four methylenes, 14 aromatic carbons with five protonated, one oxygenated sp^3^ methine, and a conjugated ketone carbonyl. Analysis of its ^1^H and ^13^C NMR data (Table 1 and Table 2) revealed the presence of a 1,2,3-trisubstituted and a 1,2,3,4-tetrasubstituted benzene rings. The COSY correlations (Figure 2) of H_2_-3/H_2_-2 and H-6/H-7 along with the HMBC correlations from H_2_-3 (*δ*_H_ 2.84) to C-4 (*δ*_C_ 204.8), C-4a (*δ*_C_ 110.7), C-1(*δ*_C_ 125.2) and C-2 (*δ*_C_ 23.6), from H_2_-2 (*δ*_H_ 3.22) to C-8a (*δ*_C_ 131.0) and C-1, from H-6 (*δ*_H_ 7.13) to C-8 (*δ*_C_ 121.3) and C-4a, and from H-7 (*δ*_H_ 8.92) to C-5 (*δ*_C_ 161.8) and C-8a, suggested the presence of a substituted 1-tetralone moiety. The observed COSY correlations of H-1′/H_2_-2′/H_2_-3′ as well as the HMBC correlations from H-1′ (*δ*_H_ 4.80) to C-8′a (*δ*_C_ 139.7) and C-4′a (*δ*_C_ 126.1), from H_2_-3′ (*δ*_H_ 2.96, 3.15) to C-4′ (*δ*_C_ 132.9), from H_2_-2′ (*δ*_H_ 1.84, 2.06) to C-4′, from H-6′ (*δ*_H_ 8.52) to C-8′ (*δ*_C_ 123.8) and C-4′a, and from H-7′ (*δ*_H_ 7.52) to C-5′ (*δ*_C_ 128.7) and C-4′a revealed the presence of a substituted 1-butanol moiety. Furthermore, the observed HMBC correlations from H_2_-2 to C-4′ and from H-3′ to C-1 suggested that there is a C-1/C-4′ linkage. The observed HMBC correlations from H-7 to C-5′ and from H-6′ to C-8 suggested that there is a C-8/C-5′ linkage. The above data revealed that compound **1** was structurally similar to altertoxin VII [11], except for the absence of a hydroxyl group located at C-7′. The ECD curve of **1** showed negative Cotton effects (CEs) around 225 and 260 nm and positive one around 280 nm, which is very similar to that of altertoxin VII [11], suggesting their same absolute configuration, i.e., 1′*R*.

Compounds **2** and **3** were obtained as yellow powder, exhibiting two totally overlapped peaks over Rp-18 column and two independent peaks with a peak area ratio of 1:2 over a chiral column (Appendix A), suggesting that they are pair of enantiomers, which were different to each other at the absolute configuration in C-1′. Their formulas were determined as C_20_H_18_O_2_ on the basis of HRESIMS, indicating 12 degrees of unsaturation. Analysis of the 1D NMR data of **2** and **3** revealed that their planar structures are similar to that of **1**, except that the C-4 carbonyl in **1** was replaced by a methylene in **2** and **3**. This deduction was further confirmed by COSY correlations of H_2_-2/H_2_-3/H_2_-4. In order to determine the absolute configuration of the stereocenter C-1′ of compounds **2** and **3**, the ECD spectra of (*R*)-**2** and (*S*)-**2** were calculated and compared with the experimental ECD spectra of **2** and **3**. As can be seen in Figure 3, the calculated ECD spectra for (*R*)-**2** and (*S*)-**2** matched well with the experimental ECD spectra of **2** and **3**, respectively, indicating that the absolute configuration of C-1′ in compound **2** is *R* and in compound **3** is *S*.

The molecular formula of compound **4** was determined as C_21_H_20_O_2_ based on HRESIMS, indicating 12 degrees of unsaturation. The ^13^C NMR data of **4** are similar to those of **2** and **3** except for the presence of a methoxy carbon (*δ*_C_ 55.2) in **4**. In the HMBC spectrum, correlation from the protons of this carbon to C-1′ (*δ*_C_ 76.8) was observed, indicating that the OH-1′ in **2** or **3** was methylated in **4**. The remaining substructure of **4** was determined to be the same as those of **2** and **3** by detailed analysis of its 2D NMR data (Figure 2). The absolute configuration of the stereocenter C-1′ in **4** was determined to be the same as that of **2** by their similar ECD curves (Figure 3) and same sign of optical rotation values.

Compound **5** was obtained as yellow powder, and its formula was determined as C_20_H_18_O_4_ on the basis of HRESIMS data, indicating 12 degrees of unsaturation. Comprehensive analysis of the NMR data of **5** revealed that its structure is very similar to that of cladosporol I (**6**), except for the absence of the C-1′ carbonyl and the presence of a double bond which was located at C-1′ (*δ*_C_ 121.9) and C-2′ (*δ*_C_ 125.9) in **5**, as deduced from the COSY correlations of H-1′/H-2′/H-3′/H-4′ as well as the HMBC correlations from H_2_-3′ (*δ*_H_ 2.29, 2.47) to C-1′ and C-4′a (*δ*_C_ 139.4) and from H-2′ (*δ*_H_ 5.89) to C-4′ (*δ*_C_ 37.5) and C-8′a (*δ*_C_ 121.2). The absolute stereochemistry of the asymmetric C-4′ was estimated from the ECD spectrum. In the ECD spectrum of **5**, there was a strong splitting cotton effect centered at 216 nm, which should be originated from exciton chirality caused by interaction between two phenolic chromophores [12]. Thus, according to the exciton chirality rule, the absolute stereochemistry of C-4′ was determined to be *S.* In addition, the absolute configuration of the C-1 stereocenter in cladosporol I (**6**) has been determined by comparison of the experimental ECD curve with the calculated ECD curves of the two C-1 epimers (1*R*,4*S*)-**6** and (1*S*,4*S*)-**6** [8]. According the results, the calculated ECD curve for (1*S*,4*S*)-**6** showed an intense positive CE around 225 nm, which matched well with the experimental one and was absent in the calculated ECD curve for (1*R*,4*S*)-**6**. Thus, the intense positive CE around 225 nm can be used to differentiate (1*R*,4*S*)-**6** and (1*S*,4*S*)-**6**. The experimental ECD curve of **5**, with an intense positive CE around 225 nm, is very similar to that of (1*S*,4*S*)-**6**, indicating that the absolute configuration of C-1 is *S*, same as that of **6**.

All the isolated compounds were tested for QS inhibitory activity against *Chromobacterium violaceum* CV026 [6]. Compounds **1**–**6** showed obvious activities (Appendix A) and the minimum inhibitory concentration (MIC) values were finally determined to be 30, 30, 20, 30, 20 and 30 μg/well, respectively.

## 3. Experimental 

### 3.1. General Experimental Procedures

The NMR spectra were recorded with a Bruker AV-500 spectrometer (Bruker, Bremen, Germany) using TMS as an internal standard. The mass spectrometric (HRESIMS) data were acquired using an API QSTAR Pulsar mass spectrometer (Bruker, Bremen, Germany). Optical rotations were measured with a JASCO P-1020 digital polarimeter (Anton Paar, Graz, Austria). The infrared spectra were recorded on a Shimadzu UV2550 spectrophotometer (Shimadzu, Kyoto, Japan). Silica gel (60–80 and 200–300 mesh; Qingdao Haiyang Chemincal Co. Ltd., Qingdao, China) and Rp-C18 (20–45 µm; Fuji Silysia Chemical Ltd., Durham, NC, USA) were used for column chromatography. Semipreparative high-performance liquid chromatography (HPLC) equipped with octadecyl silane (ODS) column (Cosmosil ODS-A, 10 × 250 nm, 5 µm, 4 mL/min) and chiral column (CHIRALPAK IC, 4.6 × 250 nm, 5 µm, 1 mL/min) were used for purification of compounds. The solvents used for the purification of compounds, such as ethyl acetate, methanol, chloroform and methanol, were of analytical pure (Concord Technology Co. Ltd., Tianjin, China).

### 3.2. Fungus Material

The fungal strain *Cladosporium* sp. KFD33 with yellow mycelium was isolated from a blood cockle in the Haikou Bay, Hainan province, in China in August 2018. After grinding, the sample (1 g) was diluted to 10^−2^ g/mL with sterile H_2_O, 100 μL of which was spread on a PDA medium plate containing chloramphenicol as bacterial inhibitor. It was identified by its morphological characteristics and 18S rRNA gene sequences (GenBank accessing No. MN737504, Appendix A), the used primers of which were NS1 (GTAGTCATATGCTTGTCTC) and NS6 (GCATCACAGACCTGTTATTGCCTC). A reference culture of *Cladosporium* sp. KFD33 is deposited in our laboratory and which maintained at −80 °C.

### 3.3. Culture Conditions

Plugs of agar, supporting mycelial growth, were cut from solid culture medium and transferred aseptically to a 1000 mL Erlenmeyer flask, containing 300 mL liquid medium (peptone 5 g/L, yeast extract 2 g/L, glucose 20 g/L, MgSO_4_ 0.5 g/L, KH_2_PO_4_ 2 g/L, pH 6.5). The fungus was cultured under static conditions at room temperature for 30 days on the shelf of our laboratory. 

### 3.4. Extraction and Isolation

The whole culture broth (20 L) was harvested and filtered to yield the mycelium cake and liquid broth. The mycelium cake and liquid broth were extracted by EtOAc for three times, respectively. The EtOAc solution was evaporated under reduced pressure. A total of 10 g EtOAc extract was obtained. The EtOAc extract was submitted to silica gel vacuum liquid chromatography using step gradient elution with PE/EtOAc (8:1, 6:1, 4:1, 2:1, 1:1, 0:1, v/v) to obtain six fractions (Frs.1–6) based on HPLC analysis. Fr.1 (2 g) was further chromatographed on Rp-C_18_ silica gel column eluted with MeOH/H_2_O (from 10%–100%) to give three subfractions (Fr.1-1–Fr.1-3). Compound **1** (1.0 mg, t_R_ 25.9 min), **5** (1.1 mg, t_R_ 10.0 min), and **6** (5.3 mg, t_R_ 19.6 min) were obtained from Fr.1-1 (15.0 mg) by ODS chromatography eluting with MeOH-H_2_O (10%–100%) and semipreparative HPLC (40% MeOH/H_2_O, containing 0.1% Formic acid, 4.0 mL·min^−1^). The mixture of compounds **2** and **3** (2.1 mg, t_R_ 10.7 min) and **4** (1.9 mg, t_R_ 32.2 min) were obtained from Fr.1-2 (6.3 mg) by ODS chromatography eluting with MeOH/H_2_O (10%–100%) and semipreparative HPLC (65% MeOH/H_2_O, containing 0.1% Formic acid, 4.0 mL·min^−1^). Compounds **2** (0.3 mg, t_R_ 24.6 min) and **3** (0.5 mg, t_R_ 27.2 min) were further obtained from their mixture by semipreparative HPLC (99% Hexane/EtOH, 1.0 mL·min^−1^), using a chiral column.

Altertoxin VIII (**1**): Dark yellow powder; ${\left[\alpha \right]}_{D}^{25}$ −9.0 (*c* 0.1, MeOH); UV (CH_3_OH) *λ*_max_ (log*ε*): 252 (3.16) nm; 217 (2.97); ECD (CH_3_OH) *λ*_max_ (∆*ε*): 222 (−1.94) nm; 237 (−0.32) nm; 250 (−0.71) nm; 270 (0.05) nm; 322 (−0.10) nm; IR(KBr) *ν*_max_: 3418, 2924, 2856, 1625, 1415, 1027 cm^−1^; ^1^H and ^13^C NMR spectral data, Table 1 and Table 2; HRESIMS m/z 327.0985 ([M + Na]^+^ calcd 327.0992).

Altertoxin IX (**2**): Yellow powder; ${\left[\alpha \right]}_{D}^{25}$ +35.0 (*c* 0.01, MeOH); UV (CH_3_OH) *λ*_max_ (log*ε*): 264 (3.14) nm; ECD (CH_3_OH) *λ*_max_ (∆*ε*): 226 (16.08) nm; 261 (1.77) nm; 278 (−0.39) nm; 312 (2.16) nm; IR(KBr) *ν*_max_: 3437, 2925, 1569, 1415 cm^−1^; ^1^H and ^13^C NMR spectral data, Table 1 and Table 2; HRESIMS m/z 313.1186 ([M + Na]^+^ calcd 313.1199).

Altertoxin X (**3**): Yellow powder; ${\left[\alpha \right]}_{D}^{25}$ −34.0 (*c* 0.01, MeOH); UV (CH_3_OH) *λ*_max_ (log*ε*): 264 (3.14) nm; ECD (CH_3_OH) *λ*_max_ (∆*ε*): 225 (−9.23) nm; 250 (1.18) nm; 265 (1.56) nm; 276 (−0.41) nm; 312 (−0.41) nm; 324 (0.34) nm; IR(KBr) *ν*_max_: 3437, 2925, 1569, 1415 cm^−1^; ^1^H and ^13^C NMR spectral data, Table 1 and Table 2; HRESIMS m/z 313.1186 ([M + Na]^+^ calcd 313.1199).

Altertoxin XI (**4**): Yellow powder; ${\left[\alpha \right]}_{D}^{25}$ +18.0 (c 0.01, MeOH); UV (CH_3_OH) *λ*_max_ (logε): 264 (3.16) nm; ECD (CH_3_OH) *λ*_max_ (∆*ε*): 226 (3.44) nm; 238 (−0.05) nm; 267 (0.48) nm; 284 (0.06) nm; 314 (0.34) nm; 336 (−0.09) nm; IR (KBr) *ν*_max_: 3440, 2924, 2854, 1689, 1584, 1418, 1207 cm^−1^; ^1^H and ^13^C NMR spectral data, Table 1 and Table 2; HRESIMS m/z 327.1342 ([M + Na]^+^ calcd 327.1356).

Altertoxin XII (**5**): Yellow powder; ${\left[\alpha \right]}_{D}^{25}$ +61.0 (*c* 0.01, MeOH); UV (CH_3_OH) *λ*_max_ (log*ε*): 261 (3.21) nm; ECD (CH_3_OH) *λ*_max_ (∆*ε*): 207 (−4.32) nm; 223 (13.32) nm; 237 (2.71) nm; 256 (6.19) nm; 272 (0.18) nm; 287 (1.80) nm; 346 (−1.24) nm; IR (KBr) *ν*_max_: 3442, 2923, 2851, 1636, 1453, 1355, 1235, 807 cm^−1^; ^1^H and ^13^C NMR spectral data, Table 1 and Table 2; HRESIMS m/z 361.0856 ([M + K]^+^ calcd 361.0837).

### 3.5. QS Inhibitory Assays of C. Violaceum CV026

An overnight culture of *C. violaceum* CV026 [9] in LB broth (OD_600_ ≈ 1.0) was prepared. Then 15 mL LB agar plate were flooded with 1 mL of this culture and C6-HSL at concentration of 500 nM to prepare an agar plate as a lawn. 5-mm diameter wells were bored in the agar plate using a flame-sterilized glass tube and the compounds dissolved in DMSO were added to the wells respectively. These plates were then incubated at 30 °C for 18 h. Inhibition of QS in *C. violaceum* is manifested as the inhibition of purple pigmentation around the wells containing the compounds

## 4. Conclusions

In conclusion, we found five new altertoxins with QS inhibitory activities against *Chromobacterium violaceum* CV026 from the marine-derived fungus *Cladosporium* sp. KFD33. These compounds represent a new type of QSIs, and can be used as lead compounds for developing new QSIs drugs, which can disarm pathogens without causing resistance of bacteria.

## Figures and Tables

**Figure 1 marinedrugs-18-00067-f001:**
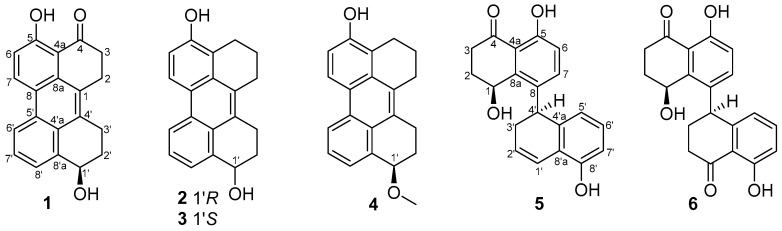
The chemical structures of compounds **1**–**6**.

**Figure 2 marinedrugs-18-00067-f002:**
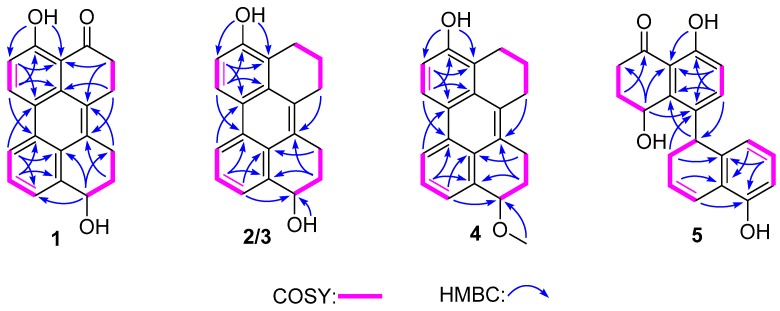
Key COSY and HMBC correlations of compounds **1**–**5**.

**Figure 3 marinedrugs-18-00067-f003:**
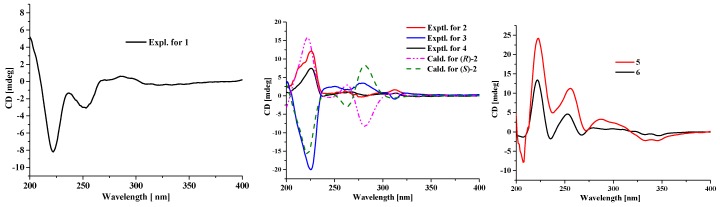
Experimental ECD spectra for compounds **1**–**6** and calculated ECD spectra for (*R*)-**2** and (S)-**2**.

**Table 1 marinedrugs-18-00067-t001:** ^1^H (500 MHz) data of **1**–**5** in DMSO.

Position	1	2/3	4	5
	*δ*_H_ (*J* in Hz)	*δ*_H_ (*J* in Hz)	*δ*_H_ (*J* in Hz)	*δ*_H_ (*J* in Hz)
1				4.81, dd, (3.2, 3.2)
2	3.22, t, (7.4)	2.84, t, (6.0)	2.92, overlap	2.01, m
				2.05, m
3	2.84, t, (7.4)	1.80, overlap	1.81, m	2.36, overlap
			1.92, overlap	2.97, ddd, (18.4, 13.2, 5.9)
4		2.78, overlap	2.78, m	
		2.86, overlap	2.90, overlap	
6	7.13, d, (9.1)	7.04, d, (9.0)	7.11, d, (9.0)	6.76, d, (8.8)
7	8.92, d, (9.1)	8.32, d, (9.0)	8.40, d, (9.0)	7.31, d, (8.8)
1′	4.80, dd, (3.6, 8.4)	4.74, dd, (8.3, 3.7)	4.47, m	6.73, overlap
2′	1.84, dtd, (13.0, 8.7, 4.7)	1.78, overlap	1.97, overlap	5.89, overlap
	2.06, m	2.00, m	2.15, m	
3′	2.96, ddd, (16.7, 9.0, 4.8)	2.82, overlap	2.89, overlap	2.29, ddd, (16.9, 6.3, 6.3)
	3.15, ddd, (16.7, 7.0, 4.8)	3.02, m	3.03, m	2.47, m
4′				4.34, dd, (12.6, 7.0)
5′				5.89, overlap
6′	8.52, d, (8.3)	8.39, d, (8.3)	8.53, d, (8.4)	6.74, overlap
7′	7.52, dd, (7.1, 8.3)	7.37, dd (7.1, 8.3)	7.43, dd, (7.0, 8.4)	6.55, d, (8.0)
8′	7.58, d, (7.1)	7.45, d, (7.1)	7.39, d, (7.0)	
9′			3.24, s	
5-OH	13.15, s		9.52, s	12.57, s
1′-OH	5.40, s	5.26, s		9.39, s

**Table 2 marinedrugs-18-00067-t002:** ^13^C NMR (125 MHz) data of **1**–**5** in DMSO.

Position	1	2/3	4	5
1	125.2, C	128.7, C	129.3, C	61.3, CH
2	23.6, CH_2_	27.1, CH_2_	27.0, CH_2_	29.8, CH_2_
3	36.3, CH_2_	21.9, CH_2_	21.8, CH_2_	32.1, CH_2_
4	204.8, C	23.4, CH_2_	23.3, CH_2_	206.3, C
4a	110.7, C	119.0, C	118.9, C	115.1, C
5	161.8, C	152.1, C	152.2, C	160.5, C
6	117.0, CH	115.5, CH	115.5, CH	117.2, CH
7	133.4, CH	121.5, CH	121.5, CH	137.3, CH
8	121.3, C	122.2, C	122.1, C	132.9, C
8a	131.0, C	129.7, C	129.6, C	142.7, C
1′	67.2, CH	67.6, CH	76.8, CH	121.9, CH
2′	31.2, CH_2_	31.5, CH_2_	27.2, CH_2_	125.9, CH
3′	24.0, CH_2_	23.8, CH_2_	22.5, CH_2_	30.3, CH_2_
4′	132.9, C	127.9, C	127.5, C	37.5, CH
4′a	126.1, C	125.9, C	126.0, C	139.4, C
5′	128.7, C	129.3, C	129.6, C	117.9, CH
6′	121.4, CH	121.2, CH	122.2, CH	127.7, CH
7′	126.2, CH	125.1, CH	124.6, CH	113.7, CH
8′	123.8, CH	123.1, CH	125.2, CH	152.6, C
8′a	139.7, C	139.2, C	134.1, C	121.2, C
9′			55.2, CH_3_

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
