# Peer review of "Altertoxins with Quorum Sensing Inhibitory Activities from The Marine-Derived Fungus Cladosporium sp. KFD33"

_marinedrugs, 2020, doi:10.3390/md18010067_

Round 1

Reviewer 1 Report

The manuscript by Zhang et al. describes physicochemical characterization and the quorum sensing inhibitory activity of sex altertoxins isolated from a marine-derived fungus. The subject is important, and the authors present adequately the methods used and the exoerimental results obtained. The manuscript would gain if more 3D-structural information could be presented (e.g. obtained by X-ray crystallography or other methods) and the functional implications (inhibitory activity) of altertoxin structures could be presented (or even speculated). The discussion part of the manuscript is very weak/ almost non-existent. The authors should discuss their results focusing on new antibiotics development and quorum sensing mechanisms. Therefore, I recommend acceptance of the manuscript only after a major revision that addresses the above issues.

Author Response

1.The manuscript would gain if more 3D-structural information could be presented (e.g. obtained by X-ray crystallography or other methods) and the functional implications (inhibitory activity) of altertoxin structures could be presented (or even speculated).

Reply: The 3D-structural information for compounds 1-5 has been added in the Supplementary Materials. We have also revised the discussion part of the manuscript as suggested, in which the functional implications of altertoxins has also been added.

2. The discussion part of the manuscript is very week/almost non-existent. The authors should discuss their results focusing on new antibiotics development and quorum sensing mechanisms.

Reply: The discussion part of the manuscript has been revised as suggested.

Reviewer 2 Report

Zhang et al, have isolated and identified five new perylenequinone derivatives, altertoxins VIII-XII, from the fermentation broth of the marine-derived Cladosporium sp. KFD33 from a blood cockle from Haikou Bay, China. All compounds showed quorum sensing inhibitory activities against Chromobacterium violaceum CV026. The planar structures and absolute configurations were appropriately confirmed.  

Few concerns before being suitable for publication;

In the introduction, need to discuss more about the QS concept and its ecological roles other than bacterial resistance, which is one of its major criteria.

In Figure 2, please remove most of unnecessary HMBC correlations for the benzene rings, especially when COSY confirmed these connections.

In the Experimental, please give more details about the fungal identification and the primers used for this purpose. In culture conditions, please be clearer about pH, shaking/static, dark/light, etc. QS inhibitory assays should be referenced.

The accurate mass of cpd 1 (HRESIMS m/z 327.0952 ([M + Na]+ calcd 327.0992), that’s more than 12 ppm difference which is not acceptable. So the authors should repeat this measurement or justify this big difference.  The same for cpd 4 (HRESIMS m/z 327.1323 ([M + Na]+  calcd 327.1356) which is around 10 ppm difference.   

Author Response

1. In the introduction, need to discuss more about the QS concept and its ecological roles other than bacterial resistance, which is one of its major criteria.

Reply: Thank you for your comments. We have added the QS concept and its ecological roles in the manuscript which are marked in red.

2. In Figure 2, please remove most of unnecessary HMBC correlations for the benzene rings, especially when COSY confirmed these connections.

Reply: According to the comments, we have removed the unnecessary HMBC correlations for the benzene rings which could confirmed by COSY correlations in Figure 2.

3. In the Experimental, please give more details about the fungal identification and the primers used for this purpose. In culture conditions, please be clearer about pH, shaking/static, dark/light, etc. QS inhibitory assays should be referenced.

Reply: We are very sorry for our negligence of the culture condition. Then, we have added primers used used, described the culture condition in detail, and QS inhibitory assays were also referenced.

4. The accurate mass of cpd 1(HRESIMS m/z 327.0952 [M + Na] + calcd 327.0992), that’s more than 12 ppm difference which is not acceptable. So, the authors should repeat this measurement or justify this big difference. The same for cpd 4 (HRESIMS m/z 327.1323 [M + Na] + calcd 327.1356) which is around 10 ppm difference.

Reply: We have resupplied the HRESIMS data of compounds 1 [HRESIMS m/z 327.0985 ([M + Na]+ calcd 327.0992)] and 4 [HRESIMS m/z 327.1342 ([M + Na]+ calcd 327.1356)].

Round 2

Reviewer 1 Report

I recommend publication of the revised version of the manuscript